# Pruning Without Fine-Tuning: Dynamic Pruning of Autoregressive Image Generation Models to Mixtures of Experts

## Abstract

While autoregressive models can achieve state-of-the-art performance in image generation, their massive size poses significant challenges for deployment and efficient model serving. Structural pruning has emerged as an effective method for reducing model size and improving inference efficiency, yet existing approaches need a recovery finetuning due to the sensitivity of image generation to missing parameters. In this work, we propose a novel approach that leverages dynamic pruning to identify and extract sparse experts within dense AR image models, enabling their transformation into Sparse Mixture of Experts (MoE) architectures. By applying top-1 expert routing to MLP layers, we establish a direct link between differentiable dynamic pruning and MoE conversion. We convert various pretrained dense models into MoEs, significantly reducing active parameters per inference step while preserving performance. Experimental results show that our approach outperforms traditional static pruning techniques by maintaining high-generation quality without costly recovery fine-tuning.

## 1 Introduction

Recent advances in image generation have emerged from two primary directions. Diffusion models (Sohl-Dickstein et al., 2015; Ho et al., 2020) excel at producing high-quality images, while there has been a resurgence of interest in AutoRegressive (AR) models (Van Den Oord et al., 2016; Van den Oord et al., 2016). The latest AR architectures closely resemble those used in modern Large Language Models (LLMs), such as Generative Pretrained Transformers (GPT) (Vaswani et al., 2017; Touvron et al., 2023; Yang et al., 2024; Abdin et al., 2024). These AR models not only demonstrate strong performance as standalone image generators (Sun et al., 2024b; Tian et al., 2025) but also integrate seamlessly into large multi-modal models, providing a unified approach to understanding and generating multi-modal content (Xie et al., 2024; Wang et al., 2024b; Chen et al., 2025).

Despite their immense capabilities, GPT models are extremely large, making deployment in resource-constrained environments highly challenging. Significant research has focused on improving the efficiency of LLMs (Frantar et al., 2022; Frantar & Alistarh, 2023; Dong et al., 2024), with structural pruning emerging as one of the most promising approaches (Ashkboos et al., 2024; Ma et al., 2023). Structural pruning removes parameters to reduce model size while achieving real wall-clock speedups, and since these techniques are designed for transformers, they can also be readily applied to transformer-based AR image generation models.

However, unlike text generation, where similar tokens could provide redundancy, image generation is highly sensitive to missing parameters. In text, a mispredicted token may still result in coherent output, but in image generation, a single incorrect token can severely degrade quality. As a result, pruned models often fail to generate high-quality or even meaningful images, necessitating a recovery fine-tuning phase to restore their generative capabilities (Ganjdanesh et al., 2024). This step is particularly problematic, as fine-tuning requires enormous computational resources, significantly more than text-based models, and obtaining high-quality image data is considerably more challenging.

One particular type of structural pruning, known as dynamic pruning (Gao et al., 2019), selects a unique subnetwork for each input, in contrast to static pruning methods, which apply the same sub-architecture to all images. Similar to dynamic pruning, Sparse Mixture of Experts (MoE)

models (Shazeer et al., 2017; Lepikhin et al., 2021) activate only a subset of parameters (the selected expert) per input. Recent studies show that MoEs can match the performance of a dense model with the same total parameters, while being far more efficient at inference (Dai et al., 2024).

However, training MoE architectures from scratch is extremely computationally demanding. Some approaches convert pretrained dense models into MoEs through continual pre-training (Zhu et al., 2024), reducing costs compared to training from scratch. Nevertheless, this continual pre-training process remains very expensive, especially for image models.

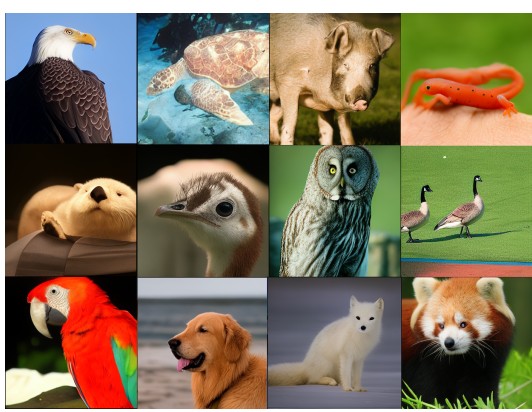

In this work, we leverage the connection between differentiable dynamic pruning and MoEs to convert a pretrained dense AR image generation model into an MoE architecture, maintaining the same total parameters but with fewer active parameters per inference step for improved efficiency. In other words, we show that experts inherently exist within dense AR image models and can be identified through dynamic pruning.

We convert the MLP layers of a pretrained AR model into MoE layers through top-1 expert routing. The routing strategy learned for dynamic structural pruning can be directly repurposed as the routing module for MoE layers. We show that unlike existing methods, our novel pruning approach preserves performance close to that of the original dense model without the

Figure 1: **Random** samples from the LlamaGen-3B model, pruned by 30% active parameters using our proposed dynamic-to-MoE pruning method, **without any recovery fine-tuning.** Despite the significant reduction in active parameters, our approach maintains strong generative performance, demonstrating that structured experts can be identified within dense AR image models without requiring continued pretraining or additional fine-tuning.

need for continued pre-training or recovery fine-tuning. A comparison of our method, static pruning, and the original model is illustrated in Fig. 2. To summarize our contributions:

1. We introduce a novel dynamic pruning method that converts pretrained dense autoregressive image generation models into sparse Mixture-of-Experts (MoE) architectures. This approach reduces the number of active parameters per inference step while maintaining performance.

2. We show experts inherently exist within dense AR image models and can be identified through dynamic structural pruning. The learned routing strategy for pruning is directly repurposed as the routing module for the MoE layers. We extend our method to compress the MLP layers via top-1 expert routing, thereby improving overall model efficiency.

3. Unlike previous methods that require continued pre-training or recovery fine-tuning to restore performance, our approach significantly outperforms the existing baselins in preserving high-quality image generation without the need for such computationally intensive steps.

4. Extensive experiments on state-of-the-art AR image generation models confirm that our method not only achieves performance close to dense models but also significantly outperforms existing pruning baselines as the number of active parameters decreases.

## 2 RELATED WORK

### 2.1 IMAGE GENERATION

High-resolution image generation is currently dominated by diffusion models (Rombach et al., 2022; Deng et al., 2009; Betker et al., 2023). While pixel-level Autoregressive (AR) image generation has been explored for years (Van den Oord et al., 2016; Chen & He, 2020), there has been a recent resurgence of interest in a new form of AR models inspired by advancements in language modeling. These models generate images by sequentially predicting the next token (Esser et al., 2021; Yu et al., 2022; Ramesh et al., 2021; Sun et al., 2024b). Typically, image tokens are derived from a pretrained discrete tokenizer, with vector quantization (VQ)(Razavi et al., 2019) or without it (Li et al., 2025).

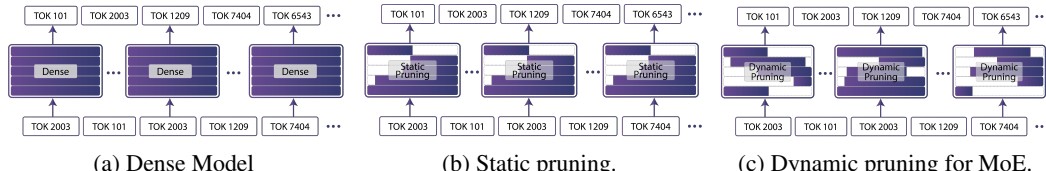

(a) Dense Model      (b) Static pruning.      (c) Dynamic pruning for MoE.

Figure 2: **(a):** The dense model uses all parameters for every input. **(b):** The statically pruned method uses the same sub-network for all tokens. **(c) Our approach** uses a different sub-network for each token utilizing MoE to constrain the expected inference budget.

## 2.2 PRUNING

Structural pruning (Li et al., 2017) shrinks models by removing unnecessary parameters without needing custom implementations. These techniques generally fall into two groups. Static pruning (Anwar et al., 2017; Molchanov et al., 2019) uses input-agnostic metrics to eliminate non-critical structures. In contrast, dynamic pruning (Gao et al., 2019; Chen et al., 2020b; Anagnostidis et al., 2023) adapts weight removal based on each input, with early work in CNNs selectively activating channels (Gao et al., 2019) and more recent efforts incorporating conditional computation in LLMs by skipping layers per token (Wang et al., 2024a). Although originally developed for LLMs, existing GPT pruning techniques (Ma et al., 2023; Frantar & Alistarh, 2023) extend naturally to language-modeling-style (next token prediction) image generation. However, they face two key hurdles: first, both static and dynamic methods degrade image generation quality to the point where prohibitive recovery fine-tuning becomes necessary; second, dynamic pruning lacks a consistent computational budget per input, which complicates batch parallelization. Our method addresses these issues by converting a dense LLM into a sparse MoE model that enforces a fixed per-token budget. Our method delivers performance close to the dense model without the need for recovery fine-tuning.

## 2.3 MIXTURE OF EXPERTS

Compared to standard structural pruning, Sparse Mixture-of-Experts (MoE) models preserve model capacity without incurring extra computational overhead. For example, Sparsely-Gated MoE (Shazeer et al., 2017) uses a learnable gating network to select a few experts per input, enabling efficient scaling to thousands of experts (Lepikhin et al., 2021). Recent methods (Dai et al., 2024) further refine expert specialization, achieving dense-model performance with a similar number of total parameters.

## 2.4 POSITIONING

Prior dense-to-MoE conversions such as LLaMA-MoE (Zhu et al., 2024) and the concurrent To-MoE (Gao et al., 2025) are designed for text LLMs and are ill-suited to AR image generation. LLaMA-MoE partitions FFN layers into experts, but due to its rigid and manual construction, it requires extensive continual pre-training to recover generation quality. This makes it prohibitive for frozen AR decoders, where fine-tuning is both computationally costly and data-limited, and where reckless parameter removal can drastically degrade output. ToMoE uses differentiable operations to learn routing over MLPs and attention heads, but its design introduces substantial architectural complexity and latency, and it fails to perform well on image generation. In contrast, our method integrates expert construction directly into the pruning stage, automatically revealing sparse structure without requiring recovery training or introducing unnecessary complexity. We specifically target AR image models because: (1) visual generation is highly sensitive to pruning, with small errors yielding visible artifacts; (2) Image models are expensive to adapt, so a post-hoc sparse conversion has high practical value; and (3) the image token setting enables interpretable expert specialization which cannot be probed in text-only models.

## 3 METHOD

The current models used to generate images are huge and computationally expensive. To reduce computational burden, we turn this dense transformer into a mixture-of-experts (MoE) framework. This involves converting the MLP block into an MoE layer with top-1 routing. Critically, unlike existing dynamic pruning methods, our approach preserves a predictable, uniform cost per token, making real-time model serving more efficient. Figure 2 provides an overview of our pruning method.

## 3.1 PRELIMINARY

Our approach builds upon decoder-only Transformer (Vaswani et al., 2017) architectures for image generation, similar to those employed in recent large-scale language models (Radford et al., 2018; Touvron et al., 2023). In the visual domain, a VQ-VAE (Razavi et al., 2019; Esser et al., 2021) is first used to convert an input image $I$ into a sequence of discrete tokens:

$$\mathbf{t} = \text{VQ-VAE}(I), \quad \mathbf{t} \in \{1, \dots, K\}^T, \quad (1)$$

where $K$ denotes the number of codes and $T$ is the token sequence length.

Once tokenized, the image is generated autoregressively by the Transformer. For each position $i$ in the sequence, the model predicts the next token based on all previous tokens:

$$\hat{t}_i = \text{Transformer}(\mathbf{t}_{<i}), \quad i = 1, \dots, T. \quad (2)$$

### 3.1.1 NOTATION

Throughout this section, we denote $T$ as the sequence length, $d$ as the hidden dimension, $d_{\text{mid}}$ as the intermediate dimension in the MLP layers. We denote the input sequence of each layer by $\mathbf{X} \in \mathbb{R}^{T \times d}$.

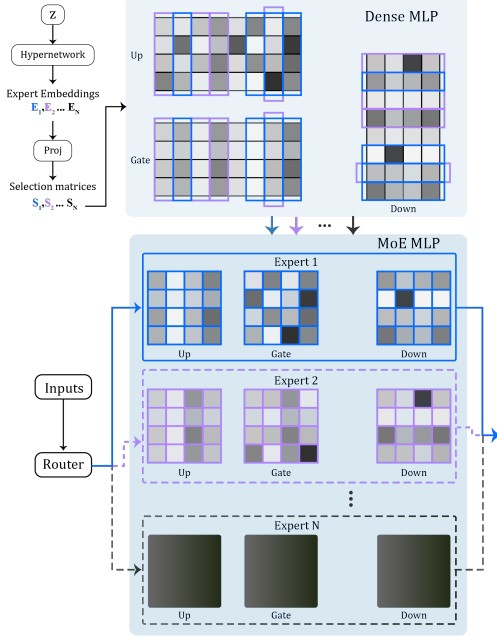

Figure 3: Overview of our pruning to MoE method. Differentiable dynamic pruning is applied to identify input-dependent experts. MLP layers are converted using top-1 expert routing.

## 3.2 EXPERT EMBEDDINGS

In our framework, a hypernetwork (Ha et al., 2016) generates learnable embeddings that guide various pruning decisions. Specifically, we sample a latent variable $\mathbf{z}$ from a fixed distribution and use it to produce an embedding matrix $\mathbf{E}_{\text{all}}$:

$$\mathbf{E}_{\text{all}} = \text{HN}(\mathbf{z}), \quad (3)$$

where $\mathbf{E}_{\text{all}} = [\mathbf{E}_1, \cdots, \mathbf{E}_l, \cdots, \mathbf{E}_L]$ contains embeddings for $L$ layers. Each individual embedding matrix $\mathbf{E}_l \in \mathbb{R}^{N \times d_e}$ represents $N$ experts, with each expert having an embedding dimension of $d_e$. These embeddings produce the subnetwork configurations in MLP experts. By having a single hypernetwork generate all embeddings, we promote knowledge sharing across layers, which enhances optimization efficiency (Gao et al., 2024).

## 3.3 MLP EXPERTS

Within a single decoder block, the MLP is expressed as

$$f(\mathbf{X}) = \sigma\Big(\mathbf{X}\,\mathbf{W}_G\Big) \odot \Big(\mathbf{X}\,\mathbf{W}_U\Big)\mathbf{W}_D, \quad (4)$$

where $\mathbf{W}_U, \mathbf{W}_G \in \mathbb{R}^{d \times d_{\text{mid}}}$ and $\mathbf{W}_D \in \mathbb{R}^{d_{\text{mid}} \times d}$ are the Up, Gate, and Down projection matrices, respectively. Here, $\sigma(\cdot)$ denotes an activation function and $\odot$ represents the element-wise product.

To improve efficiency and enable specialization, we split the MLP into $N$ experts, each corresponding to a pruned version of the intermediate dimension $d_{\text{mid}}$. When a subset of tokens $\mathbf{X}_t$ are routed to expert $i$, its function becomes

$$f^i(\mathbf{X}_t) = \sigma\Big(\mathbf{X}_t\,\mathbf{W}_G\,\mathbf{S}_i\Big) \odot \Big(\mathbf{X}_t\,\mathbf{W}_U\,\mathbf{S}_i\Big)\mathbf{S}_i^\top\,\mathbf{W}_D, \quad (5)$$

where $\mathbf{S}_i = \text{Diag}(\mathbf{s}_i)$ is a diagonal mask of dimension $\mathbb{R}^{d_{\text{mid}} \times d_{\text{mid}}}$. Each entry in $\mathbf{s}_i$ is either 0 or 1, indicating whether a neuron is pruned or retained. We jointly learn the routing decisions and the mask values through

$$\mathbf{s} = \text{ST}-\text{GS}\Big(\text{Proj}_\text{D}\big(\mathbf{G}\,\mathbf{E}\big)\Big), \mathbf{G} = \text{ST}-\text{G}\Big(\text{Router}(\mathbf{X})\Big). \quad (6)$$

Here, $\mathbf{E}$ (omitting layer indices for brevity) is an embedding from the hypernetwork, and $\mathrm{Proj}_{\mathrm{D}} : \mathbb{R}^{d_e} \to \mathbb{R}^{d_{\mathrm{mid}}}$ is a linear projection that maps this embedding into the MLP's intermediate space. The operators $\mathrm{ST-GS}$ and $\mathrm{ST-G}$ denote the straight-through Gumbel-Sigmoid and Gumbel-Softmax, respectively (Jang et al., 2016), that enable differentiable routing among experts. Given a logit vector $\mathbf{x}$, we first draw noise from the standard Gumbel distribution $\mathbf{g} \sim \mathrm{Gumbel}(0,1)$. We then form a probability distribution via

$$\mathbf{p} = \mathrm{softmax}\left(\frac{\mathbf{x} + \mathbf{g}}{\tau}\right), \tag{7}$$

where $\tau$ is a temperature parameter that regulates the sharpness of the distribution. To allow discrete expert selection during training while preserving gradient flow, we employ a straight-through estimator. This is accomplished by replacing $\mathbf{p}$ with a one-hot vector corresponding to the maximum value:

$$\mathrm{ST-G}(\mathbf{x}) = \text{one-hot}\left(\arg\max_i \frac{x_i + g_i}{\tau}\right). \tag{8}$$

For binary decisions, we similarly use a sigmoid function in place of softmax. An additive bias $b$ is incorporated to ensure all experts are initially active:

$$\mathrm{ST-GS}(\mathbf{x}) = \mathrm{round}(\mathrm{sigmoid}(\frac{\mathbf{x} + \mathbf{g} + b}{\tau})) \tag{9}$$

### 3.4 EFFECTIVE MoE REGULARIZATION

#### 3.4.1 UNION OF EXPERTS

To preserve the full expressiveness of the original dense model, we encourage the union of active parameters from all experts to cover the complete set of neurons. Let

$$\mathbf{u} = \bigcup_{i=1}^{N} \mathbf{s}_i \quad \Longrightarrow \quad \mathbf{u} = 1 - \prod_{i=1}^{N}(1 - \mathbf{s}_i), \tag{10}$$

so that an element of $\mathbf{u}$ is active if any expert retains that neuron. We then encourage the proportion $\frac{\sum \mathbf{u}}{|\mathbf{u}|}$ to go toward 1:

$$\mathcal{R}_{\mathrm{U}} = \frac{1}{L}\sum_{l=1}^{L} f\left(\frac{\sum \mathbf{u}_l}{|\mathbf{u}_l|}, 1\right). \tag{11}$$

$f(x,y) = \log\left[\max(x,y)/\min(x,y)\right]$ measures the divergence between the achieved and desired activation levels. This ensures that, together, the experts maintain the full capacity of the dense model.

#### 3.4.2 PARAMETER BUDGET

We further enforce an upper bound on the total number of active parameters. For each layer $l$ in the MLP, we define the maximum expert width as

$$d_l^* = \max(\mathbf{s}\, \mathbf{1}_{d_{\mathrm{mid}}}), \tag{12}$$

where $d_{\mathrm{mid}}$ is the expanded width of the MLP layer, and aggregate these widths into

$$\mathbf{d}_{\mathrm{MoE}} = [d_1^*, \ldots, d_L^*]. \tag{13}$$

We then compare the total active parameters, $\mathrm{T}(\mathbf{d}_{\mathrm{MoE}})$, with the full model's total $\mathrm{T}_{\mathrm{total}}$ scaled by a target ratio $p \in (0,1]$:

$$\mathcal{R}_{\mathrm{P}} = f\left(\mathrm{T}(\mathbf{d}_{\mathrm{MoE}}), p\,\mathrm{T}_{\mathrm{total}}\right). \tag{14}$$

This constraint controls the overall model size, ensuring parameter efficiency.

#### 3.4.3 LOAD BALANCING

To avoid overloading a few experts while underutilizing others, we adopt a load balancing loss similar to that of the Switch Transformer (Fedus et al., 2022). Let $F_i$ denote the fraction of tokens routed to expert $i$ and $P_i$ the average softmax probability (prior to straight-through sampling) for expert $i$. Then, the load balancing loss is given by

$$\mathcal{R}_{\mathrm{L}} = N \sum_{i=1}^{N} F_i\, P_i. \tag{15}$$

This term encourages an even distribution of tokens among the experts, promoting balanced utilization.

## 3.5 TRAINING THE MoE CONSTRUCTION

To learn our MoE, we freeze the original decoder weights and train only the router, hypernetwork, and projection parameters. The final model thus retains the knowledge of the dense precursor while gaining MoE capabilities (Figure 3).

We define the training objective:

$$
\min_{\theta} \mathcal{L}\big(\underbrace{f'(\mathbf{X}; \mathbf{E}_{\text{all}})}_{\text{MoE model}}, \ \underbrace{f(\mathbf{X})}_{\text{dense model}}\big) + \alpha\,\mathcal{R}_{\text{P}} + \beta\,\mathcal{R}_{\text{U}} + \gamma\,\mathcal{R}_{\text{L}}, \tag{16}
$$

where $\theta = [\theta_{\text{HN}}, \theta_{\text{Router}}, \theta_{\text{Proj}_{\text{D}}}]$, and $\mathcal{L}$ is the sum of distillation loss (Hinton et al., 2015) between the dense model $f$ and the MoE model $f'$ and language modeling loss. We employ in-place knowledge distillation to guide the sparse model without additional memory cost (Muralidharan et al., 2024).

At the end of training, each MLP layer is replaced with $N$ experts sharing weights. We also support switching back to a pseudo-MoE version, which may simplify distributed training. In summary, our approach provides a flexible means of transforming a dense autoregressive image decoder into a sparse mixture-of-experts model, greatly reducing computational load while retaining the dense model's capability. See Appendix B for more details of our method.

## 4 EXPERIMENTS

### 4.1 EXPERIMENTAL SETTINGS

**Models:** We evaluate our method on the class-conditional autoregressive models LlamaGen-XXL and LlamaGen-3B (Sun et al., 2024b), both trained on ImageNet (Deng et al., 2009). Additionally, we consider Janus-Pro-7B (Chen et al., 2025), an autoregressive multi-modal model capable of both image understanding and generation.

**Baselines:** We compare our approach against state-of-the-art static and dynamic structural pruning methods, including LLM-Pruner (Ma et al., 2023), SliceGPT (Ashkboos et al., 2024), SLEB (Song et al., 2024), and DISP-LLM (Gao et al., 2024), all of which have official implementations available.

**Datasets:** For training our hypernetwork, we use ImageNet (Russakovsky et al., 2015) for LlamaGen models and the COCO (Lin et al., 2014) 2017 training set for Janus-Pro. If a baseline requires a dataset for pruning or gradient calculations, we use the same datasets.

**Evaluation:** For evaluating class-conditional models, we report FID (Heusel et al., 2017), Inception Score (Salimans et al., 2016), sFID (Nash et al., 2021), and Precision/Recall (Kynkäänniemi et al., 2019) on 5000 samples from the ImageNet 2012 validation set. Following LlamaGen (Sun et al., 2024b), We generate the images at $384 \times 384$ and resize them to $256 \times 256$ for evaluation. We do not use top-k decoding unless specified otherwise, and the CFG (Ho & Salimans, 2022) scale is set to 1.5 for all models as it is the default value in the released Llamagen codebase.

| Method | Model/Speed | | ImageNet (256×256) Quality | | |
| --- | --- | --- | --- | --- | --- |
| | Params (B) | Latency (s/it) | IS ↑ | FID ↓ | Precision ↑ |
| No Pruning | | | | | |
| LlamaGen-XXL | 1.411 | 17.20 | 207.89 | 15.58 | 0.7992 |
| Sparsity = 0.2 | | | | | |
| LLM-Pruner | 1.145 | 13.24 | 49.15 | 42.86 | 0.4530 |
| SLEB | 1.156 | 12.00 | 75.28 | 32.60 | 0.5904 |
| Slice-GPT | 1.140 | 12.96 | 10.41 | 104.62 | 0.2450 |
| DISP-LLM | 1.138 | 18.97 | 148.58 | 17.88 | 0.7310 |
| **Ours (MoE)** | 1.131A / 1.399T | 12.06 | **159.10** | **17.31** | **0.7500** |
| Sparsity = 0.3 | | | | | |
| LLM-Pruner | 0.994 | 12.21 | 23.24 | 65.77 | 0.2674 |
| SLEB | 1.014 | 10.59 | 17.33 | 91.17 | 0.2734 |
| Slice-GPT | 0.983 | 11.58 | 7.80 | 122.36 | 0.1940 |
| DISP-LLM | 1.004 | 18.88 | 90.40 | 23.58 | 0.6636 |
| **Ours (MoE)** | 0.989A / 1.231T | 11.37 | **135.33** | **18.56** | **0.7276** |

Table 1: Comparison for LlamaGen-XXL on ImageNet (5k val, 256×256): model size, latency, and quality metrics across pruning ratios, **without recovery finetuning**. Full Table in Appendix C.3.

**Implementation Details:** We prune LlamaGen-XXL (Sun et al., 2024b) using our method and the baselines at two sparsity levels: 0.2 and 0.3. LlamaGen-3B (Sun et al., 2024b) and Janus-Pro-7B (Chen et al., 2025) are pruned at three levels: 0.3, 0.4, and 0.5. We set the loss weighting parameters to $\alpha, \beta, \gamma = (16.0, 2.0, 1.0)$ (Eq. 16). The model weights remain frozen while we train our hypernetwork for 10,000 iterations. We then convert the model to a MoE using the hypernetwork

and evaluate it. No recovery fine-tuning is performed on either the baselines or our method. For all experimental details see Appendix C.

## 4.2 RESULTS

### 4.2.1 CLASS-CONDITIONAL RESULTS

First, we evaluate the effectiveness of our method on the moderately sized LlamaGen-XXL (Sun et al., 2024b) model, which has 1.4B parameters. Table 1 presents the results. Notably, the gap between our method with no recovery finetuning and the dense model is minimal across key metrics, including FID, sFID, Precision, and Recall.

Another important observation is the sensitivity of image generation models to pruning. All baseline methods, except for DISP-LLM (Gao et al., 2024), perform poorly, failing to generate coherent images, as reflected in their IS and FID scores, as well as class-wise precision and recall. This further supports our argument that image generation models are more sensitive to parameter removal compared to text generation models and need a recovery finetuning. Interestingly, these same baselines perform reasonably well in the context of LLM pruning.

Furthermore, at the same active parameter ratio, our method consistently outperforms all baselines across all metrics by a significant margin. With a pruning rate of 30%, our method surpasses LLM-Pruner (Ma et al., 2023), SliceGPT (Ashkboos et al., 2024), and SLEB (Song et al., 2024), even when these baselines retain more active parameters (with only a 20% pruning rate). While our method and DISP-LLM (Gao et al., 2024) show comparable performance at the 20% pruning level, our advantage becomes more pronounced as the number of active parameters decreases. Specifically, at 70% active parameters, our method achieves a 50% higher IS and approximately 20% lower FID, demonstrating its superior robustness to pruning.

Next, we evaluate our method and the baselines on the larger LlamaGen-3B (Sun et al., 2024b) model, as shown in Table 2. Again, our method performs very close to the dense model at a 30% sparsity rate and achieves higher recall at 70% and 60% active parameters, demonstrating that its effectiveness is not limited to smaller models and performs just as well, if not better, on larger models.

The performance gap between our method and the best baseline, i.e. DISP-LLM (Gao et al., 2024), is even larger than in the LlamaGen-XXL case, particularly at lower pruning ratios. Our method achieves 70% higher IS and 28% lower FID. This gap widens significantly as the number of active parameters decreases: at 60% active parameters, we outperform the best baseline by 272% in IS and 53% in FID, and at 50% active parameters, by 158% in IS and 40% in FID. Meanwhile, we again observe the generation capabilities of the other baselines degrade severely without fine-tuning.

| Method | Model/Speed | | ImageNet (256×256) Quality | | |
| | Params (B) | Latency (s/it) | IS ↑ | FID ↓ | Precision ↑ |
|---|---|---|---|---|---|
| **No Pruning** | | | | | |
| LlamaGen-3B | 3.097 | 13.75 | 240.33 | 16.05 | 0.8232 |
| **Sparsity = 0.3** | | | | | |
| LLM-Pruner | 2.174 | 10.81 | 22.49 | 71.46 | 0.2984 |
| SLEB | 2.225 | 9.81 | 26.04 | 66.91 | 0.3268 |
| Slice-GPT | 2.178 | 10.87 | 13.74 | 90.54 | 0.3624 |
| DISP-LLM | 2.198 | 12.77 | 101.89 | 22.76 | 0.7022 |
| **Ours (MoE)** | 2.167A / 2.998T | 10.32 | **172.19** | **16.37** | **0.7800** |
| **Sparsity = 0.4** | | | | | |
| LLM-Pruner | 1.867 | 10.30 | 11.14 | 103.38 | 0.1482 |
| SLEB | 1.976 | 9.72 | 8.27 | 130.30 | 0.1830 |
| Slice-GPT | 1.855 | 10.51 | 10.27 | 111.41 | 0.2686 |
| DISP-LLM | 1.901 | 12.70 | 32.84 | 42.94 | 0.5420 |
| **Ours (MoE)** | 1.858A / 2.820T | 10.21 | **122.23** | **20.06** | **0.7096** |
| **Sparsity = 0.5** | | | | | |
| LLM-Pruner | 1.559 | 9.95 | 8.89 | 120.67 | 0.1048 |
| SLEB | 1.602 | 9.24 | 4.77 | 171.31 | 0.1218 |
| Slice-GPT | 1.531 | 10.07 | 6.78 | 135.21 | 0.2076 |
| DISP-LLM | 1.604 | 12.46 | 19.08 | 62.63 | 0.4448 |
| **Ours (MoE)** | 1.548A / 2.239T | 9.98 | **49.11** | **37.59** | **0.5652** |

Table 2: Comparison for LlamaGen-3B on ImageNet (5k val, 256×256): model size, latency, and quality metrics across pruning ratios, **without recovery finetuning**. Full Table in Appendix C.3.

### 4.2.2 TEXT-CONDITIONAL RESULTS

We also apply our method and the baselines to the Janus-Pro-7B (Chen et al., 2025) model to evaluate their effectiveness on a much larger, text-conditional model. The results are presented in Table 9. For evaluating the model, we report FID (Heusel et al., 2017) and CLIP Score (Hessel et al., 2021) on 5000 samples from the COCO 2017 validation set, also resized to 256 × 256. Additionally, we report PickScore (Kirstain et al., 2023) on the PartiPrompts (Yu et al., 2022) as a proxy for human preference. We do not use top-k decoding and set the CFG weight to 5.0 (the default in Janus codebase).

First, we observe that the dense model struggles with image generation, as evidenced by its high FID score of 58.69. Even in this setting, our method remains competitive with the dense model at a lower pruning ratio of 30% and significantly outperforms the baselines. Janus uses the VQ-VAE from LlamaGen (Sun et al., 2024b) as its tokenizer for image generation, but both text and image tokens are processed by the same transformer backbone. Consequently, the hyper-network and router modules for the MLP experts in our method must be trained on both token types. Given the relatively short nature of the COCO captions, we hypothesize that the COCO training set may be too small to fully support the router module's demands. We believe that a larger dataset would enable our method to perform even better in this scenario. Furthermore, based on prior findings in (He et al., 2024), we suspect that converting the model into two distinct MLP mixtures-of-experts (one for text and one for images) could further enhance our method's effectiveness. This represents an intriguing direction for future work.

| Method | Model/Speed | | Quality | | |
|---|---|---|---|---|---|
| | Params (B) | Latency (s/it) | FID ↓ | CLIP Score ↑ | PickScore ↑ |
| No Pruning | | | | | |
| Janus-Pro-7B | 6.910 | 10.66 | 58.69 | 28.25 | 19.5637 |
| Sparsity = 0.3 | | | | | |
| LLM-Pruner | 4.868 | 9.92 | 129.67 | 21.57 | 18.2333 |
| SLEB | 5.089 | 7.46 | 278.56 | 19.45 | 18.1239 |
| Slice-GPT | 4.920 | 10.05 | 194.36 | 20.42 | 18.2444 |
| DISP-LLM | 5.101 | 11.02 | 122.61 | 22.09 | 18.7462 |
| **Ours (MoE)** | 4.845A / 6.688T | 9.86 | **71.55** | **26.31** | **19.4181** |
| Sparsity = 0.4 | | | | | |
| LLM-Pruner | 4.154 | 9.28 | 179.33 | 19.74 | 17.9993 |
| SLEB | 4.481 | 6.51 | 284.52 | 19.36 | 18.1144 |
| Slice-GPT | 4.162 | 9.21 | 213.36 | 20.32 | 17.9082 |
| DISP-LLM | 4.492 | 10.95 | 151.83 | 21.65 | 18.2357 |
| **Ours (MoE)** | 4.146A / 5.964T | 9.32 | **104.56** | **24.99** | **18.6615** |
| Sparsity = 0.5 | | | | | |
| LLM-Pruner | 3.501 | 9.15 | 184.36 | 19.06 | 17.8421 |
| SLEB | 3.874 | 6.36 | 300.47 | 19.31 | 18.0559 |
| Slice-GPT | 3.496 | 8.53 | 339.87 | 19.25 | 17.7116 |
| DISP-LLM | 3.885 | 10.88 | 188.14 | 20.62 | 17.8908 |
| **Ours (MoE)** | 3.485A / 4.416T | 8.99 | **126.41** | **22.83** | **18.1969** |

Table 3: Comparison on Janus-Pro-7B for text-conditional generation: COCO-2017 (256×256) FID and CLIP score, and PickScore on PartiPrompts, **without recovery finetuning**.

We have also reported parameter counts and average latency measurements over 5 generated images for all our experiments. For the reported latency values, our memory usage correlates with the total params and remains lower than that of the base model. Even at a sparsity rate of 50%, our method outperforms all baselines across all sparsity rates (except for Disp-LLM at 30%). Notably, at 50% sparsity, our method achieves similar total parameter counts and latency to the baselines at 30%, but with significantly better performance. Our approach outperforms Disp-LLM at comparable sparsity rates while being much faster as Disp-LLM incurs additional overhead due to index selection and addition operations. We control for compute by matching budgets across models (Appendix C.3.1).

### 4.2.3 QUALITATIVE COMPARISONS

Figure 1 shows some image generations from the LlamaGen-3B (Sun et al., 2024b) model, pruned to a MoE with 70% active parameters using our method, without any recovery fine-tuning. Our model is able to generate high-quality images without the need for a fine-tuning phase. We provide more generated samples of our method and a qualitative comparison with the baselines in Appendix C.4.1.

### 4.2.4 EXPERT ASSIGNMENT RESULTS

To better understand how the LlamaGen-3B (Sun et al., 2024b) MoE model with 70% active parameters processes information, we analyze token routing across different experts to gain insights into expert specialization, class-wise token distribution, and spatial token assignments within input image tokens. As shown in Figure 4, expert selection in the first layer (Figure 4b) appears balanced, with no clear spatial pattern in token routing. However, by layer 11 (Figure 4c), a spatial pattern seems to emerge.

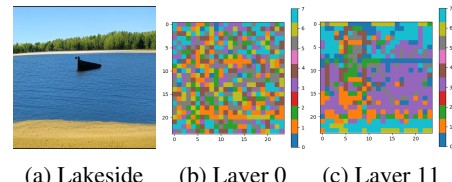

(a) Lakeside    (b) Layer 0    (c) Layer 11

Figure 4: Token routing for two layers of the LlamaGen-3B-MoE.

To gain higher-level insights into potential spatial relationships in expert routing, we generate 1,000 images from the model and analyze which tokens, corresponding to different parts of the image, are processed by each expert at different layers. Figure 5 presents the results for layers 11 and 19. This figure clearly shows that certain experts exhibit spatial specialization, tending to process specific regions of the image. We believe this

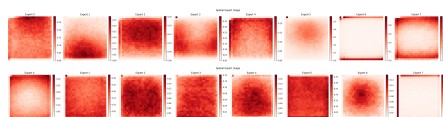

Figure 5: Spatial token assignments across experts of LlamaGen-3B-MoE (Top: Layer 11, Bottom: Layer 19). Darker red indicates more routed tokens.

finding aligns with previous observations in MoE LLMs(Jiang et al., 2024), where experts are found to specialize in syntax of text rather than semantics.

Figure 6 illustrates how token distribution varies across different ImageNet classes. Certain experts are predominantly activated for specific categories, indicating that the model learns some class-specialized expert assignments.

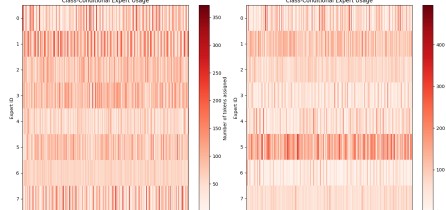

Figure 6: Class-wise distribution of token routing across experts in LlamaGen-3B-MoE (Left: Layer 13, Right: Layer 19).

### 4.3 COMPARISON TO OTHER DENSE-TO-MOE METHODS

Table 4 compares our method to LLaMA-MoE Zhu et al. (2024) and ToMoE Gao et al. (2025) when converting LlamaGen-3B to 60% active parameters.

LLaMA-MoE and ToMoE both suffer from higher latency due to added architectural complexity and less efficient routing. LLaMA-MoE requires substantial continued pre-training to restore performance, so we report results under two conditions: zero-shot (no retraining similar to our method) and with 3k steps of full fine-tuning. For ToMoE, we replicate its 10k-step calibration phase. In contrast, our method requires no continued

| Method | Params(B) | Latency ↓ | FID ↓ |
|---|---|---|---|
| Llama-MoE | 1.760A / 3.097T | 12.09 | 136.21 |
| Llama-MoE (Tuned) | 1.760A / 3.097T | 12.09 | 48.53 |
| ToMoE | 1.863A / 2.907T | 13.61 | 39.71 |
| Ours | 1.858A / 2.820T | 10.21 | **20.06** |

Table 4: Comparison of our method with dense to moe baselines on LLmagen-3B.

pre-training, keeps the dense backbone entirely frozen, is less complex, and still achieves lower latency and superior FID compared to both baselines. These results highlight that integrating expert construction directly into the pruning phase yields a sparse MoE that is both deployment-friendly and robust, even in the brittle setting of autoregressive image generation.

### 4.4 ABLATION STUDY

We conduct an additional experiment to study the impact of the various components of our method when pruning LlamaGen-3B (Sun et al., 2024b) to 70% active parameters. We begin with a simple hypernetwork regularized using only the parameter and union regularization losses (Eq. 11 and Eq. 14). We then incrementally add the other components.

| Method | IS ↑ | FID ↓ | Prec. ↑ |
|---|---|---|---|
| HN + Union Loss (Eq. 11) | 140.88 | 18.59 | 0.7290 |
| + Load Balance Loss (Eq. 15) | 154.85 | 18.03 | 0.7476 |
| + Distillation Loss | 174.73 | 16.35 | 0.7792 |
| + Language Modeling Loss | 172.19 | 16.37 | 0.7800 |

Table 5: Ablations on 5k ImageNet validation set.

Table 5 presents the results. We observe that incorporating the load balance loss (Eq. 15) improves the results by ensuring that all experts are assigned an adequate number of tokens. Furthermore, adding the distillation loss further enhances performance. While the inclusion of the language modeling loss (where the hypernetwork is supervised with actual image tokens rather than the outputs of the teacher networks) slightly impacts the generation quality, it notably increases precision and recall, as expected when training with real images. Consequently, we report our results using a combination of both distillation and language modeling losses in Table 1, Table 2, and Table 3 and despite observing slightly poorer FID and IS values. Overall, Table 5 highlights the importance of each component of our method. See Appendix C.4 for more ablations.

## 5 CONCLUSION

We introduced a dynamic pruning approach that transforms dense autoregressive image generation models into efficient Sparse Mixture of Experts architectures. By leveraging top-1 expert routing, our method extracts specialized experts, significantly reducing active parameters per inference step while preserving high image quality without the need for recovery fine-tuning. Our evaluations show that this strategy outperforms conventional pruning techniques and maintains robust performance even at high sparsity levels. Furthermore, analysis of expert routing reveals inherent spatial and class-specific specialization. Overall, our work provides a scalable solution for efficient deployment of large AR models in resource-constrained environments.

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

# A   RELATED WORK

## A.1   IMAGE GENERATION

High-resolution image generation is currently dominated by diffusion models (Rombach et al., 2022; Deng et al., 2009; Ramesh et al., 2022; Betker et al., 2023). While pixel-level Autoregressive (AR) image generation has been explored for years (Gregor et al., 2014; Van Den Oord et al., 2016; Van den Oord et al., 2016; Chen et al., 2020a; Parmar et al., 2018), there has been a recent resurgence of interest in a new form of AR models inspired by advancements in language modeling. These models generate images by sequentially predicting the next token (Esser et al., 2021; Yu et al., 2022; Ramesh et al., 2021; Sun et al., 2024b; Ramesh et al., 2022; Ding et al., 2022; Lu et al., 2022) or the next token map (Tian et al., 2025). Typically, image tokens are derived from a pretrained discrete tokenizer, where a finite vocabulary is obtained via vector quantization (VQ)(Razavi et al., 2019). However, some approaches have explored autoregressive image generation without relying on vector quantization (Li et al., 2025)

## A.2   PRUNING

Structural pruning (Li et al., 2017; Kurtic et al., 2022; Ma et al., 2023) offers a practical way to shrink models by removing unnecessary parameters without needing custom implementations. These techniques generally fall into two groups. Static pruning (Anwar et al., 2017; Molchanov et al., 2019; Fang et al., 2023) uses input-agnostic metrics to eliminate non-critical structures. In contrast, dynamic pruning (Gao et al., 2019; Chen et al., 2020b; Anagnostidis et al., 2023; Dong et al., 2024) adapts weight removal based on each input, with early work in CNNs selectively activating channels (Gao et al., 2019; Chen et al., 2020b) and more recent efforts incorporating conditional computation in LLMs by skipping layers per token (Wang et al., 2024a). Although originally developed for LLMs, existing GPT pruning techniques (Ma et al., 2023; Ashkboos et al., 2024; Song et al., 2024; van der Ouderaa et al., 2024; Gao et al., 2024; Lin et al., 2024; Men et al., 2024; Frantar & Alistarh, 2023; Sun et al., 2024a) extend naturally to language-modeling-style (next token prediction) image generation. However, they face two key hurdles: first, both static and dynamic methods degrade image generation quality to the point where prohibitive recovery fine-tuning becomes necessary; second, dynamic pruning lacks a consistent computational budget per input, which complicates batch parallelization. Our method addresses these issues by converting a dense LLM into a sparse MoE model that enforces a fixed per-token budget. Our method delivers performance close to the dense model without the need for recovery fine-tuning.

## A.3   MIXTURE OF EXPERTS

Compared to standard structural pruning, Sparse Mixture-of-Experts (MoE) models preserve model capacity without incurring extra computational overhead. For example, Sparsely-Gated MoE (Shazeer et al., 2017) uses a learnable gating network to select a few experts per input, enabling efficient scaling to thousands of experts (Lepikhin et al., 2021). Recent methods (Dai et al., 2024) further refine expert specialization, achieving dense-model performance with a similar number of total parameters.

# B   METHOD

## B.1   MODULE ARCHITECTURE

Tab. 6 summarizes our design. In our approach, each MLP layer is equipped with its own adapter and router modules. After training, only the router module is retained.

### B.1.1   EMBEDDING GENERATION

The hypernetwork receives a fixed random vector $z \in \mathbb{R}^{N \times 32}$, sampled from a normal distribution, and produces an embedding. In parallel, the token features are mapped into a 128-dimensional space via the MLP Adapter ($\theta_{\text{Proj}_D}$). For a network with $L$ MLP layers, each layer is equipped with its own MLP Adapter and Expert Router. During training of the hypernetwork, we use Eq. 5 and 6 to find the expert embeddings and routing decision. After training, we prune the MLP layers to N experts

| Component | Removed? | Config |
|-----------|----------|--------|
| Hypernetwork | Yes | Random vector $z \rightarrow$ BiGRU(32, 64) |
| MLP Adapter | Yes | LayerNorm(128) $\rightarrow$ GeLU $\rightarrow$ Linear(128, $d_e$) |
| Expert Router | No | Linear($d$, $N$) |

Table 6: Configurations of the trainable components. Components marked for removal are pruned after training while their outputs are preserved for expert generation.

using the hypernetwork output expert embeddings. The we remove the hypernetwork and the adapter module and use the router module as the MoE router.

## C EXPERIMENTS

### C.1 DETAILED EXPERIMENTAL SETTINGS

We prune LlamaGen-XXL Sun et al. (2024b) using our method and baseline approaches at sparsity levels of 0.2 and 0.3. LlamaGen-3B Sun et al. (2024b) and Janus-Pro-7B Chen et al. (2025) are pruned at sparsity levels of 0.3, 0.4, and 0.5. The loss weighting parameters are set to $\alpha, \beta, \gamma = (16.0, 2.0, 1.0)$(Eq. 16) and the distillation and language modeling losses are weighted equally at 1.0. We set $\tau = 0.4$ in Eq. 8 and 9 and $b = 3.0$ in Eq. 8 and 9. The model weights remain frozen while the hypernetwork is trained for 10,000 iterations. Afterward, we convert the model to a MoE using the trained hypernetwork and evaluate it without any recovery fine-tuning applied to either the baselines or our method. We use the AdamW optimizer with a constant learning rate of 0.0004, weight decay of 0.05, and Adam parameters $(\beta_1, \beta_2) = (0.9, 0.999)$. The batch size is set to 1. For the LlamaGen Sun et al. (2024b) models, we use a single NVIDIA A6000 GPU, while for Janus-Pro-7B Chen et al. (2025), we use a single Nvidia H100 GPU.

### C.2 MORE RESULTS

### C.3 CLASS CONDITIONAL RESULTS

First, we evaluate the effectiveness of our method on the moderately sized LlamaGen-XXL (Sun et al., 2024b) model, which has 1.4B parameters. Table 7 presents the results. Notably, the gap between our method with no recovery finetuning and the dense model is minimal across key metrics, including FID, sFID, Precision, and Recall.

Another important observation is the sensitivity of image generation models to pruning. All baseline methods, except for DISP-LLM (Gao et al., 2024), perform poorly, failing to generate coherent images, as reflected in their IS and FID scores, as well as class-wise precision and recall. This further supports our argument that image generation models are more sensitive to parameter removal compared to text generation models and need a recovery finetuning. Interestingly, these same baselines perform reasonably well in the context of LLM pruning.

Furthermore, at the same active parameter ratio, our method consistently outperforms all baselines across all metrics by a significant margin. With a pruning rate of 30%, our method surpasses LLM-Pruner (Ma et al., 2023), SliceGPT (Ashkboos et al., 2024), and SLEB (Song et al., 2024), even when these baselines retain more active parameters (with only a 20% pruning rate). While our method and DISP-LLM (Gao et al., 2024) show comparable performance at the 20% pruning level, our advantage becomes more pronounced as the number of active parameters decreases. Specifically, at 70% active parameters, our method achieves a 50% higher IS and approximately 20% lower FID, demonstrating its superior robustness to pruning.

Next, we evaluate our method and the baselines on the larger LlamaGen-3B (Sun et al., 2024b) model, as shown in Table 8. Again, our method performs very close to the dense model at a 30% sparsity rate and achieves higher recall at 70% and 60% active parameters, demonstrating that its effectiveness is not limited to smaller models and performs just as well, if not better, on larger models.

The performance gap between our method and the best baseline, i.e. DISP-LLM (Gao et al., 2024), is even larger than in the LlamaGen-XXL case, particularly at lower pruning ratios. Our method

| Method | Model/Speed | | ImageNet (256×256) Quality | | | | |
| | Params (B) | Latency (s/it) | IS ↑ | FID ↓ | sFID ↓ | Precision ↑ | Recall ↑ |
| --- | --- | --- | --- | --- | --- | --- | --- |
| **No Pruning** | | | | | | | |
| LlamaGen-XXL (1.4B) | 1.411 | 17.20 | 207.89 | 15.58 | 73.80 | 0.7992 | 0.7484 |
| **Sparsity = 0.2** | | | | | | | |
| LLM-Pruner | 1.145 | 13.24 | 49.15 | 42.86 | 77.72 | 0.4530 | 0.7298 |
| SLEB | 1.156 | 12.00 | 75.28 | 32.60 | 80.13 | 0.5904 | 0.7302 |
| Slice-GPT | 1.140 | 12.96 | 10.41 | 104.62 | 91.27 | 0.2450 | 0.3672 |
| DISP-LLM | 1.138 | 18.97 | 148.58 | 17.88 | 75.50 | 0.7310 | 0.7358 |
| **Ours (MoE)** | 1.131 / 1.399 | 12.06 | **159.10** | **17.31** | **75.14** | **0.7500** | **0.7524** |
| **Sparsity = 0.3** | | | | | | | |
| LLM-Pruner | 0.994 | 12.21 | 23.24 | 65.77 | 82.06 | 0.2674 | 0.6608 |
| SLEB | 1.014 | 10.59 | 17.33 | 91.17 | 97.67 | 0.2734 | 0.5492 |
| Slice-GPT | 0.983 | 11.58 | 7.80 | 122.36 | 98.08 | 0.1940 | 0.3146 |
| DISP-LLM | 1.004 | 18.88 | 90.40 | 23.58 | 77.98 | 0.6636 | 0.7312 |
| **Ours (MoE)** | 0.989 / 1.231 | 11.37 | **135.33** | **18.56** | **75.92** | **0.7276** | **0.7460** |

Table 7: Comparison for LlamaGen-XXL on ImageNet (5k val, 256×256): model size, latency, and quality metrics across pruning ratios, **without recovery finetuning**.

achieves 70% higher IS and 28% lower FID. This gap widens significantly as the number of active parameters decreases: at 60% active parameters, we outperform the best baseline by 272% in IS and 53% in FID, and at 50% active parameters, by 158% in IS and 40% in FID. Meanwhile, we again observe the generation capabilities of the other baselines degrade severely without fine-tuning. See Appendix C.3.2 for results on Janus Pro.

We have also reported parameter counts and average latency measurements over 5 generated images for all our experiments. For the reported latency values, our memory usage correlates with the total params and remains lower than that of the base model. Even at a sparsity rate of 50%, our method outperforms all baselines across all sparsity rates (except for Disp-LLM at 30%). Notably, at 50% sparsity, our method achieves similar total parameter counts and latency to the baselines at 30%, but with significantly better performance. Our approach outperforms Disp-LLM at comparable sparsity rates while being much faster as Disp-LLM incurs additional overhead due to index selection and addition operations.

### C.3.1 COMPUTE DISCUSSION

As emphasized throughout the paper, fine-tuning the base model after pruning is often prohibitively expensive in terms of memory, compute, and data requirements, especially for large-scale models. A core assumption in our work is that such post-pruning fine-tuning is infeasible in many practical scenarios, and our method is explicitly designed to avoid this costly step. Our approach only optimizes a tiny router (See Table 10) on frozen model weights, which can be efficiently trained for models as large as 11B on a consumer 24GB GPU. Other baselines also require *calibration* phases and we have accounted to have just about the same compute when feasible: **The strongest baseline DISP-LLM** also trains a hypernetwork for 10k iterations. This is why we use 10k iterations with a batch size of 1. **SLEB** and **SliceGPT** perform multiple forward passes to decide which blocks or weights to prune, e.g. at 0.5 sparsity on Janus, we allocate 8,280 base model forward passes for SLEB and 8,192 iterations for SliceGPT. This is practical as long as no gradients of the base model are required. But **LLM-Pruner** requires computing gradients for all model parameters. Notably, we could not apply LLM-Pruner to Janus-7B on the RTX6000-48GB GPUs used in our other experiments and had to resort to H100-80GB GPUs, illustrating that when you have to compute gradients (for calibration or FT), GPU memory demands are significantly higher.

### C.3.2 TEXT-CONDITIONAL RESULTS

We also apply our method and the baselines to the recently released Janus-Pro-7B (Chen et al., 2025) model to evaluate their effectiveness on a much larger, text-conditional model. The results are presented in Table 9. For evaluating the model, we report FID (Heusel et al., 2017) and CLIP Score (Hessel et al., 2021) on 5000 samples from the COCO 2017 validation set, also resized to

| Method | Model/Speed | | ImageNet (256×256) Quality | | | | |
| --- | --- | --- | --- | --- | --- | --- | --- |
| | Params (B) | Latency (s/it) | IS ↑ | FID ↓ | sFID ↓ | Precision ↑ | Recall ↑ |
| **No Pruning** | | | | | | | |
| LlamaGen-3B | 3.097 | 13.75 | 240.33 | 16.05 | 73.23 | 0.8232 | 0.7300 |
| **Sparsity = 0.3** | | | | | | | |
| LLM-Pruner | 2.174 | 10.81 | 22.49 | 71.46 | 87.67 | 0.2984 | 0.6134 |
| SLEB | 2.225 | 9.81 | 26.04 | 66.91 | 82.03 | 0.3268 | 0.6604 |
| Slice-GPT | 2.178 | 10.87 | 13.74 | 90.54 | 98.35 | 0.3624 | 0.3958 |
| DISP-LLM | 2.198 | 12.77 | 101.89 | 22.76 | 80.49 | 0.7022 | 0.7052 |
| **Ours (MoE)** | 2.167 / 2.998 | 10.32 | **172.19** | **16.37** | **76.64** | **0.7800** | **0.7464** |
| **Sparsity = 0.4** | | | | | | | |
| LLM-Pruner | 1.867 | 10.30 | 11.14 | 103.38 | 96.60 | 0.1482 | 0.4784 |
| SLEB | 1.976 | 9.72 | 8.27 | 130.30 | 103.04 | 0.1830 | 0.3548 |
| Slice-GPT | 1.855 | 10.51 | 10.27 | 111.41 | 105.48 | 0.2686 | 0.3460 |
| DISP-LLM | 1.901 | 12.70 | 32.84 | 42.94 | 82.18 | 0.5420 | 0.6238 |
| **Ours (MoE)** | 1.858 / 2.820 | 10.21 | **122.23** | **20.06** | **77.50** | **0.7096** | **0.7434** |
| **Sparsity = 0.5** | | | | | | | |
| LLM-Pruner | 1.559 | 9.95 | 8.89 | 120.67 | 109.84 | 0.1048 | 0.2944 |
| SLEB | 1.602 | 9.24 | 4.77 | 171.31 | 119.86 | 0.1218 | 0.0220 |
| Slice-GPT | 1.531 | 10.07 | 6.78 | 135.21 | 112.87 | 0.2076 | 0.3080 |
| DISP-LLM | 1.604 | 12.46 | 19.08 | 62.63 | 87.61 | 0.4448 | 0.5314 |
| **Ours (MoE)** | 1.548 / 2.239 | 9.98 | **49.11** | **37.59** | **82.45** | **0.5652** | **0.6750** |

Table 8: Comparison for LlamaGen-3B on ImageNet (5k val, 256×256): params, latency, and quality metrics across pruning ratios, **without recovery finetuning**.

256 × 256. Additionally, we report PickScore (Kirstain et al., 2023) on the PartiPrompts (Yu et al., 2022) as a proxy for human preference. We do not use top-k decoding and set the CFG weight to 5.0 (the default in Janus codebase).

First, we observe that the dense model struggles with image generation, as evidenced by its high FID score of 58.69. Even in this setting, our method remains competitive with the dense model at a lower pruning ratio of 30% and significantly outperforms the baselines. Janus uses the VQ-VAE from LlamaGen (Sun et al., 2024b) as its tokenizer for image generation, but both text and image tokens are processed by the same transformer backbone. Consequently, the hypernetwork and router modules for the MLP experts in our method must be trained on both token types. Given the relatively short nature of the COCO captions, we hypothesize that the COCO training set may be too small to fully support the router module's demands. We believe that a larger dataset would enable our method to perform even better in this scenario. Furthermore, based on prior findings in (He et al., 2024), we suspect that converting the model into two distinct MLP mixtures-of-experts—one for text and one for images—could further enhance our method's effectiveness. This represents an intriguing direction for future work.

### C.3.3 PARAMETER COUNT AND LATENCY

Base model finetuning is substantially more memory, compute and data intensive and often infeasible for large models. In contrast, our approach only optimizes a tiny router (See Tab. 10) on frozen model weights, which can be efficiently trained for models as large as 11B on a consumer 24GB GPU. Other baselines also require *calibration* phases and we have accounted to have just about the same compute when feasible: **The strongest baseline DISP-LLM** also trains a hypernetwork for 10k iterations. This is why we use 10k iterations with a batch size of 1. **SLEB** and **SliceGPT** perform multiple forward passes to decide which blocks or weights to prune, e.g. at 0.5 sparsity on Janus, we allocate 8,280 base model forward passes for SLEB and 8,192 iterations for SliceGPT. This is practical as long as no gradients of the base model are required. But **LLM-Pruner** requires computing gradients for all model parameters. Notably, we could not apply LLM-Pruner to Janus-7B on the RTX6000-48GB GPUs used in our other experiments and had to resort to H100-80GB GPUs, illustrating that when you have to compute gradients (for calibration or FT), GPU memory demands are significantly higher.

| Method | Model/Speed | | Quality | | |
| --- | --- | --- | --- | --- | --- |
| | Params (B) | Latency (s/it) | COCO FID ↓ | CLIP Score ↑ | PickScore ↑ |
| **No Pruning** | | | | | |
| Dense (Janus-Pro-7B) | 6.910 | 10.66 | 58.69 | 28.25 | 19.5637 |
| **Sparsity = 0.3** | | | | | |
| LLM-Pruner | 4.868 | 9.92 | 129.67 | 21.57 | 18.2333 |
| SLEB | 5.089 | 7.46 | 278.56 | 19.45 | 18.1239 |
| Slice-GPT | 4.920 | 10.05 | 194.36 | 20.42 | 18.2444 |
| DISP-LLM | 5.101 | 11.02 | 122.61 | 22.09 | 18.7462 |
| **Ours (MoE)** | 4.845 / 6.688 | 9.86 | **71.55** | **26.31** | **19.4181** |
| **Sparsity = 0.4** | | | | | |
| LLM-Pruner | 4.154 | 9.28 | 179.33 | 19.74 | 17.9993 |
| SLEB | 4.481 | 6.51 | 284.52 | 19.36 | 18.1144 |
| Slice-GPT | 4.162 | 9.21 | 213.36 | 20.32 | 17.9082 |
| DISP-LLM | 4.492 | 10.95 | 151.83 | 21.65 | 18.2357 |
| **Ours (MoE)** | 4.146 / 5.964 | 9.32 | **104.56** | **24.99** | **18.6615** |
| **Sparsity = 0.5** | | | | | |
| LLM-Pruner | 3.501 | 9.15 | 184.36 | 19.06 | 17.8421 |
| SLEB | 3.874 | 6.36 | 300.47 | 19.31 | 18.0559 |
| Slice-GPT | 3.496 | 8.53 | 339.87 | 19.25 | 17.7116 |
| DISP-LLM | 3.885 | 10.88 | 188.14 | 20.62 | 17.8908 |
| **Ours (MoE)** | 3.485 / 4.416 | 8.99 | **126.41** | **22.83** | **18.1969** |

Table 9: Comparison on Janus-Pro-7B for text-conditional generation: COCO-2017 (256×256) FID and CLIP score, and PickScore on PartiPrompts, **without recovery finetuning**.

| Model | Base Size (B) | HN Size (B) | Pct (%) |
| --- | --- | --- | --- |
| LG-XXL | 1.41 | 0.02 | 1.8% |
| LG-3B | 3.10 | 0.03 | 0.9% |
| Janus-7B | 6.91 | 0.04 | 0.6% |

Table 10: Router vs Base Model size.

## C.4 MORE ABLATIONS

Table 11 shows the full ablation results. In another ablation experiment, we investigate how the number of experts affects generation results. Figure 7b illustrates a uncorrelated relationship between IS and FID, where improvements in one do not necessarily lead to improvements in the other. Overall, increasing the number of experts from 8 to 12 improves generation quality in terms of FID; however, the improvements are minor and may not justify the added complexity. Moreover, further increasing the number of experts from 12 to 16 results in poorer quality, possibly because the hypernetwork and router module cannot be sufficiently trained with so many experts.

Finally, we visualize the expert width and the union of experts width across all layers of the model in Figure 7a. We can see that there is a trend and latter layers have higher width compared to initial layers. Also our method does a good job of forcing the union of experts to be close to the dense model (Eq. 11).

| Method | IS ↑ | FID ↓ | sFID ↓ | Prec. ↑ | Rec. ↑ |
| --- | --- | --- | --- | --- | --- |
| HN + Union Loss (Eq. 11) | 140.88 | 18.59 | 76.99 | 0.7290 | 0.7452 |
| + Load Balance Loss (Eq. 15) | 154.85 | 18.03 | 77.50 | 0.7476 | 0.7420 |
| + Distillation Loss | 174.73 | 16.35 | 76.05 | 0.7792 | 0.7392 |
| + Language Modeling Loss | 172.19 | 16.37 | 76.64 | 0.7800 | 0.7464 |

Table 11: Ablations on 5k ImageNet validation set.

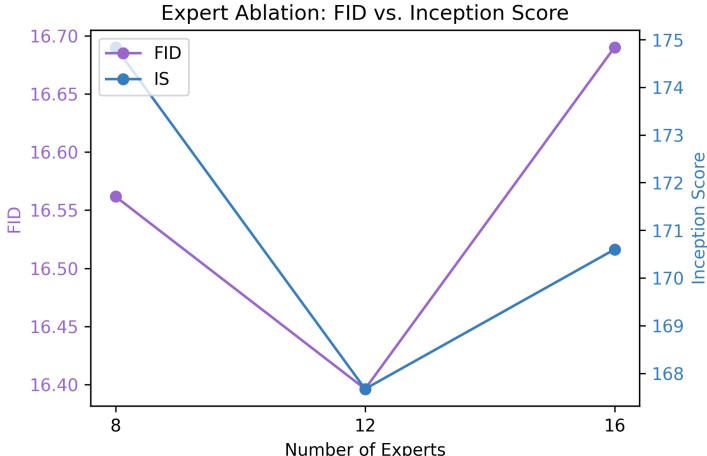

(a) Normalized expert width and union of experts across LlamaGen-3B MLP layers.

(b) Effect of number of experts on generation results of LlamaGen-3B-MoE with 70% active parameters.

Figure 7: LlamaGen-3B MoE analysis: (a) expert width/union across MLP layers; (b) impact of number of experts at 70% active parameters.

### C.4.1 MORE GENERATIONS

In this section we present more visual results of our method as well some MoE expert analysis results. Figure 8 compares our dynamic-to-MoE pruning method with baseline approaches on the LlamaGen-3B(Sun et al., 2024b) model, pruned to 70% active parameters. Our method not only surpasses all baselines by a large margin in visual fidelity but, in some cases (second and fifth images), even produces better images than the dense model. Both Figure 1 and Figure 8 were generated using a CFG scale of 1.5 and a top-k value of 200. Fig. 9 depicts more randomly sampled generations of the LlamaGen-3B Sun et al. (2024b) model pruned to 70% active parameters using our method.

Fig. 10 illustrates how token distribution varies across different ImageNet classes. Certain experts are predominantly activated for specific categories, indicating that the model learns some class-specialized expert assignments. This pattern is not visible in all layers.

To gain higher-level insights into potential spatial relationships in expert routing, we generate 1,000 images from the model and analyze which tokens, corresponding to different parts of the image, are processed by each expert at different layers. Fig. 11 and 12 present the results for all layers. This figure clearly shows that certain experts exhibit spatial specialization, tending to process specific regions of the image. We believe this finding aligns with previous observations in MoE LLMsJiang et al. (2024), where experts are found to specialize in syntax of text rather than semantics.

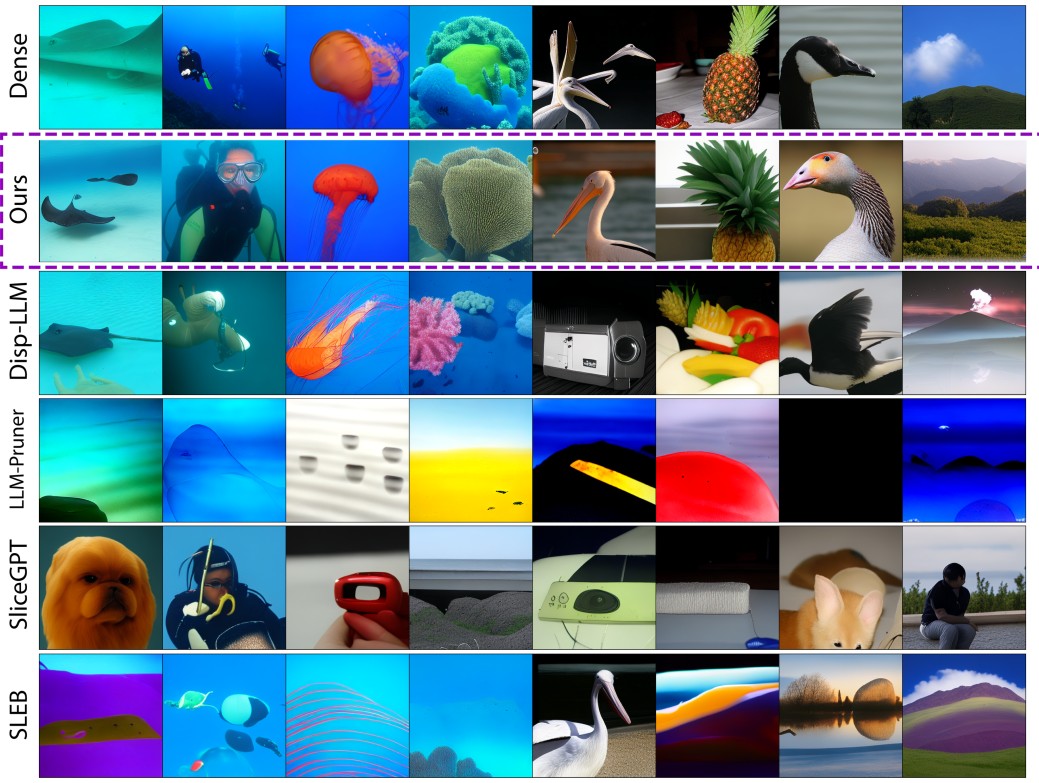

Figure 8: Samples from the LlamaGen-3B model, pruned by 30% of active parameters using our proposed method, compared to baseline methods **without recovery fine-tuning**. Our approach significantly outperforms the baselines in visual fidelity.

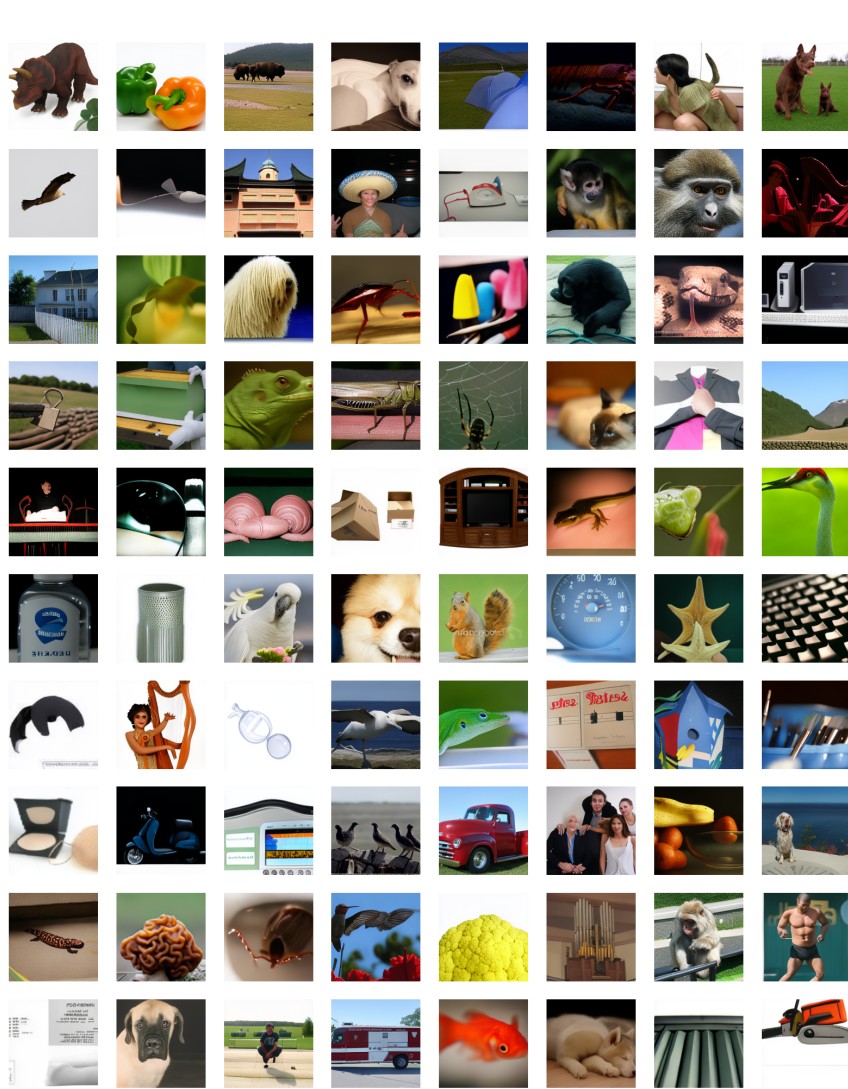

Figure 9: **Random** samples from the LlamaGen-3B model, pruned by 30% active parameters using our proposed dynamic-to-MoE pruning method, **without any recovery fine-tuning.** Despite the significant reduction in active parameters, our approach maintains strong generative performance, demonstrating that structured experts can be identified within dense AR image models without requiring continued pretraining or additional fine-tuning.

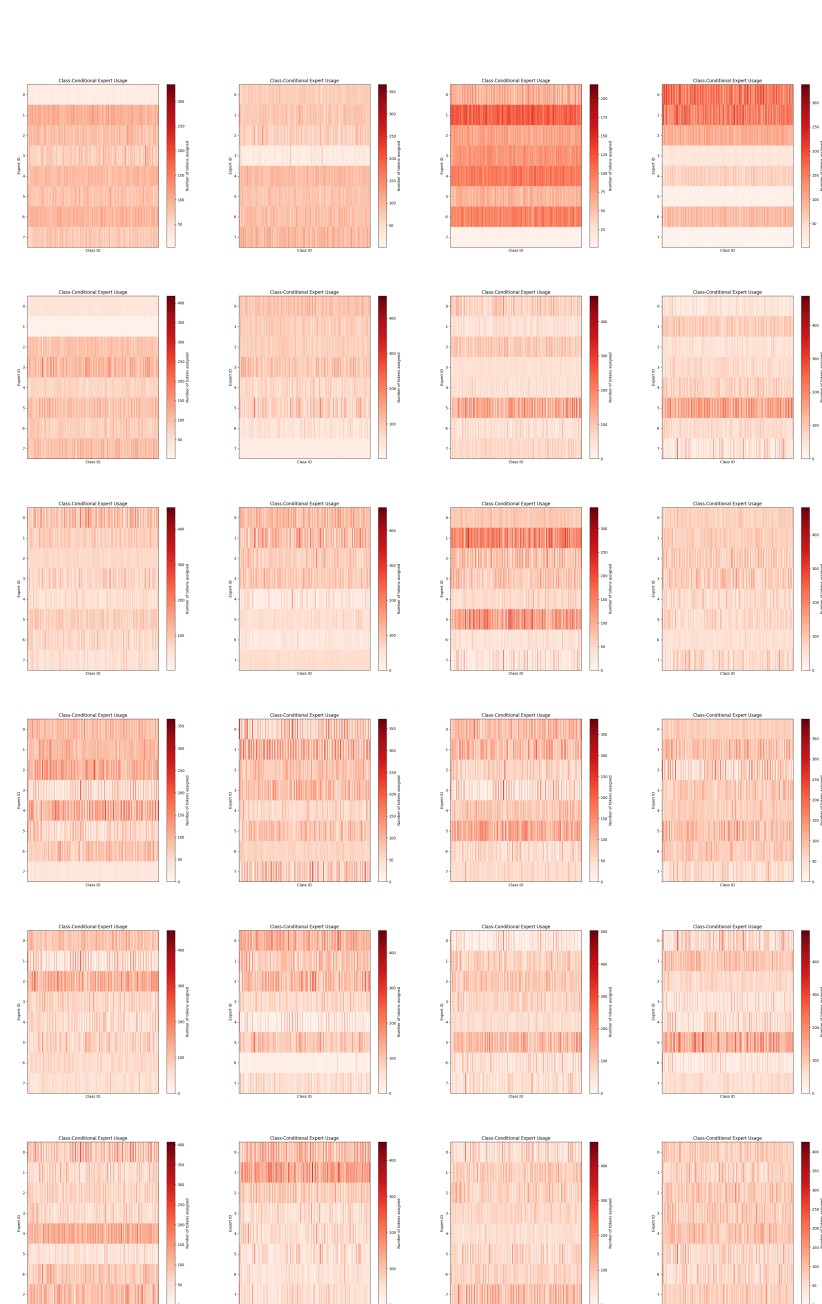

Figure 10: Class-wise distribution of token routing across experts in LlamaGen-3B-MoE, showing class specialized expert assignment. Each bar represents the proportion of tokens assigned to different experts for a given class, showing that certain experts are preferentially activated for specific categories. Layers 0–23 are displayed from the top-left to the bottom-right.

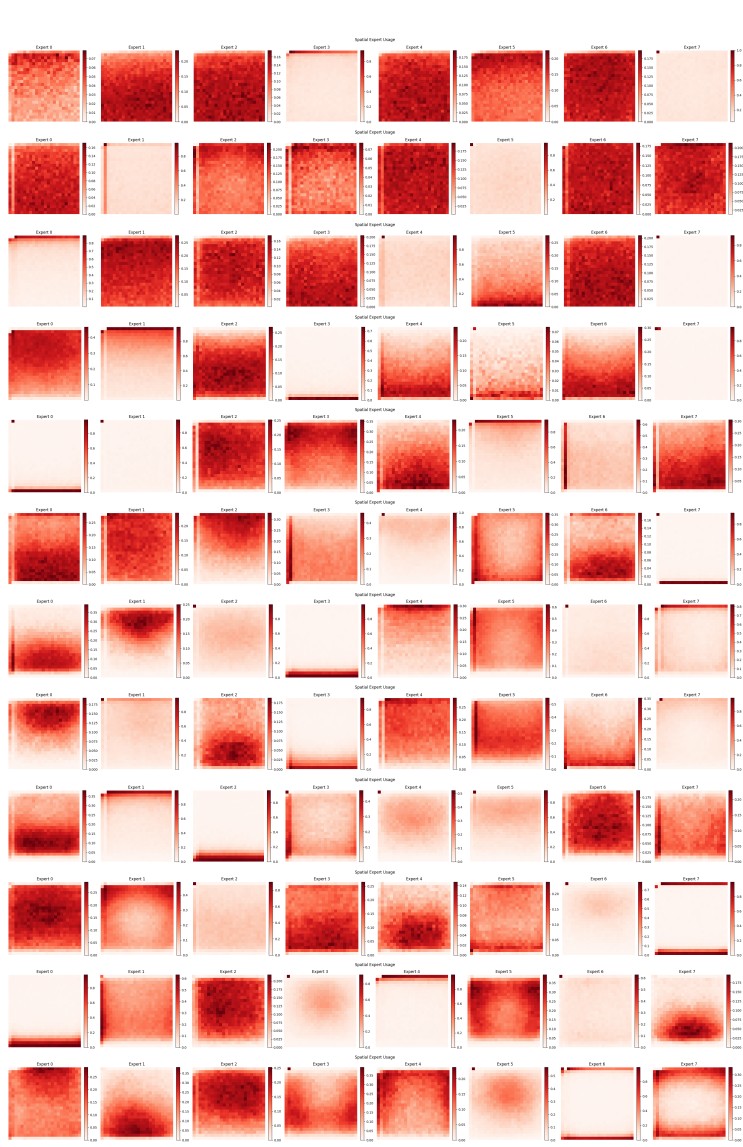

Figure 11: Spatial token assignments across experts of LlamaGen-3B-MoE. Darker red indicates a higher number of routed tokens. Layers 0–11 are displayed from top to bottom.

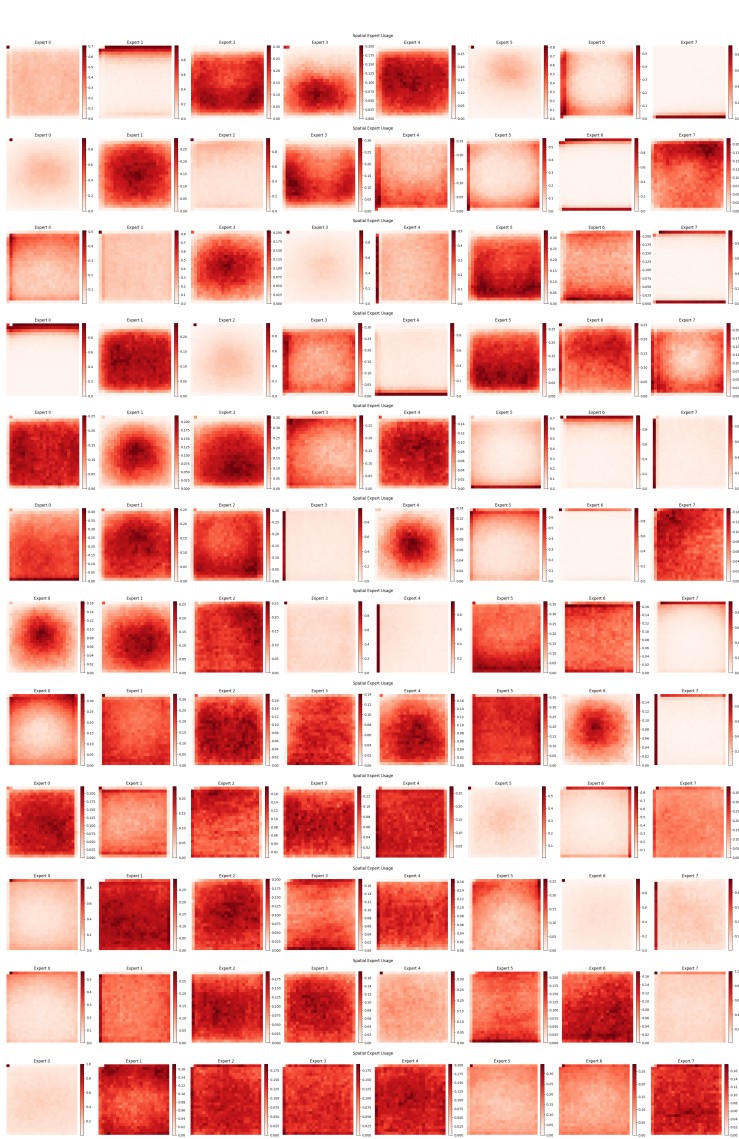

Figure 12: Spatial token assignments across experts of LlamaGen-3B-MoE. Darker red indicates a higher number of routed tokens. Layers 12–23 are displayed from top to bottom.

