# OpenReview forum: "Pruning Without Fine-Tuning: Dynamic Pruning of Autoregressive Image Generation Models to Mixtures of Experts"
_ICLR.cc/2026/Conference — ICLR 2026 Conference Withdrawn Submission_

### Official Review · Reviewer_fN4p · 2025-10-29

**Soundness:** 2
**Presentation:** 2
**Contribution:** 2
**Rating:** 2
**Confidence:** 5

**Summary:**

This paper proposes a method to dynamically prune the Auto-Regressive (AR) image generation models with the idea of Mixture-of-Experts (MoE). That is to say, take the sparse model as the MoE.

The authors reduce the number of activated (used) parameters during the image generation, the activated parameters are dynamically selected for each inference step.

The experimental results show that, without fine-tuning, the proposed method achieves better performance than several baseline methods.

**Strengths:**

1. This paper achieves better performance than other baselines.

2. This paper introduce the MoE with sparsity to build the efficient image generation model.

**Weaknesses:**

1. The experiments reported in the paper is not fair. All baselines (LLM-Pruner, SLEB, Slice-GPT, DISP-LLM) are designed for Large Language Models (LLMs) as textual tasks rather than visual (image) generation tasks. Thus, it is unfair to compare to those baselines.

2. The work [1] claims his proposed structural pruning method is effective for both textual and visual generation tasks, but the authors did not compare their method to this work.

3. The whole architecture of the revised model is similar as the MoE format, the novelty is limited.

4. The claim of 'Pruning without finetuning' is misleading as the router and hypernetwork are still trained on the same dataset (ImageNet and CoCo). The same training effort should be applied to other baseline methods on the same dataset for fair comparison.

5. As the fine-tuning is still there, the work may compare the proposed method to the LoRA fine-tuning on structure pruned models.

---
[1] Numerical Pruning for Efficient Autoregressive Models

**Questions:**

1. Is there any comparison to other pruning methods designed for image generation?

2. Experiments with same training efforts for other baseline methods?

3. Compare to LoRA?

---

### Official Review · Reviewer_jV79 · 2025-10-29

**Soundness:** 2
**Presentation:** 2
**Contribution:** 2
**Rating:** 2
**Confidence:** 4

**Summary:**

This paper proposes an approach that leverages dynamic pruning to identify and extract sparse experts within dense AR image models,
enabling their transformation into Sparse Mixture of Experts (MoE) architectures. By applying top-1 expert routing to MLP layers, it establishes a direct link between differentiable dynamic pruning and MoE conversion. It convert the MLP layers of a pretrained AR
model into MoE layers through top-1 expert routing. The routing strategy learned for dynamic structural pruning can be directly repurposed as the routing module for MoE layers.

**Strengths:**

It converts various pretrained dense models into MoEs, reducing active parameters per inference step while preserving performance.

It leverages the connection between differentiable dynamic pruning and MoEs to convert a pretrained dense AR image generation model into an MoE architecture, maintaining the same total parameters but with fewer active parameters per inference step for improved efficiency. In other words, it shows that experts inherently exist within dense AR image models and can be identified through dynamic pruning.

**Weaknesses:**

The novelty may be limited. The technical components such as MoE layers, Gumbel-Sigmoid, and  Gumbel-Softmax are commonly adopted in MoE or pruning works. The technical contribution may be limited.

It claims that it does not need finetuning in the title and paper. But it still needs to train additional networks including the router, hypernetwork, and projection parameters. At the end of training, each MLP layer is replaced with N experts sharing weights. In actual, the model architecture is changed and part of the model is still finetuned. It may be be accurate to claim without finetuning.

The comparison may not be fair. The proposed method needs retraining. The baselines typically needs to finetune the model with some text data. But for image generation, how to finetune the baselines from text language domain with what kind of data, it is still an open question. The paper does not provide much details about how the baselines are finetuned. It may not be fair to compare to baselines without any finetuning while the proposed method goes through finetuning. It is better to provide more details.

The paper focus on image generation. But all baselines are from text language domain. It addresses decoder-only Transformer architectures like typical LLMs. The proposed method can be seamlessly applied to LLMs. It is better to report and compare with the baselins for typical LLM models. It would be a bit strange if all methods and baselines are for LLMs, but the experimental results are for image generation only.

From the ablation study in table 5, the most simple version with only HN + Union Loss, can achieve 140 IS, 18 FID, 0.729 Prec, which is already better than all baselines in  Table 2. It seems that the baselines are not properly configured or finetuned with enough data, so that their performance are so weak, worse than the proposed method even without distillation loss or language modeling loss. The most simple version with only HN + Union Loss, does not seem to incorporate any loss related to the generation performance. It is mainly related with parameters since the union loss encourages the union of active parameters from all experts to cover the complete set of neurons. It is better to discuss why the performance is better.

**Questions:**

see the weakness.

---

### Official Review · Reviewer_uLLk · 2025-10-30

**Soundness:** 2
**Presentation:** 2
**Contribution:** 1
**Rating:** 2
**Confidence:** 4

**Summary:**

This paper presents Pruning Without Fine-Tuning, a novel dynamic pruning framework for autoregressive (AR) image generation models that directly converts dense architectures into Sparse Mixture-of-Experts (MoE) models without any recovery fine-tuning. The core idea is to reinterpret differentiable dynamic pruning as top-1 expert routing in MLP layers, revealing latent expert structures within pretrained AR models such as LlamaGen-3B/XXL and Janus-Pro-7B. The method introduces a hypernetwork-guided pruning mechanism that learns input-dependent neuron masks using Gumbel-Softmax and Gumbel-Sigmoid sampling, coupled with regularization terms to preserve model expressivity (union loss), control parameter budget, and balance expert utilization.
Empirical results on ImageNet and COCO datasets demonstrate that the proposed approach outperforms existing static and dynamic pruning baselines (e.g., LLM-Pruner, SliceGPT, SLEB, DISP-LLM) in both efficiency and generative quality, maintaining near-dense performance even at 50–70% sparsity without any fine-tuning

**Strengths:**

Introduces the first method to convert dense AR image models into sparse MoE architectures without retraining, a breakthrough for deployment efficiency.

well-written

**Weaknesses:**

Limited novelty: The method essentially applies dynamic pruning and MoE conversion techniques from LLM research to the AR image generation setting. The theoretical framework and training pipeline are nearly identical to prior dense-to-MoE approaches (e.g., ToMoE 2025, LLaMA-MoE 2024).

Lack of broader evaluation: The experiments are restricted to ImageNet and COCO, missing more challenging or standardized text-conditional generation benchmarks such as Geneval, PartiPrompts, or VBench, which would better demonstrate robustness and semantic fidelity.

No new architectural insight: The proposed model retains the same transformer backbone and MLP design, without introducing new structures or routing strategies specific to visual autoregression.

**Questions:**

See Weaknesses below

---

### Official Review · Reviewer_C61s · 2025-11-01

**Soundness:** 3
**Presentation:** 3
**Contribution:** 3
**Rating:** 6
**Confidence:** 5

**Summary:**

In this work, a novel approach that leverages dynamic pruning to identify and extract sparse experts within dense AR image models, enabling their transformation into Sparse Mixture of Experts (MoE) architectures.

**Strengths:**

see Summary

**Weaknesses:**

The authors should provide additional supplementary materials, including at least 30 to 50 visual results to better demonstrate the capabilities and limitations of their method. These visual results should include both qualitative and quantitative comparisons with other methods.

The paper mentions a user study but lacks detailed results and analysis. User studies are crucial for understanding the perceptual quality and practical usability of the generated videos.

The authors should conduct additional experiments on text-to-image generation using more recent benchmarks. Specifically, the authors should test their method on several newer text-to-image benchmarks to demonstrate its effectiveness in this domain.

**Questions:**

no

---

### Note · Authors · 2025-11-12

I have read and agree with the venue's withdrawal policy on behalf of myself and my co-authors.